# Cardiovascular Evaluation of Liver Transplant Patients by Using Coronary Calcium Scoring in ECG-Synchronized Computed Tomographic Scans

**DOI:** 10.3390/jcm10215148

**Published:** 2021-11-02

**Authors:** Anna Bettina Roehl, Marc Hein, Johanna Kroencke, Felix Kork, Alexander Koch, Anne Andert, Michael Becker, Jonas Schmöe, Sebastian Daniel Reinartz

**Affiliations:** 1Department of Anesthesiology, Medical Faculty, RWTH Aachen University, 52074 Aachen, Germany; mhein@ukaachen.de (M.H.); jkroencke@ukaachen.de (J.K.); fkork@ukaachen.de (F.K.); 2Department of Internal Medicine III, Gastroenterology, Metabolic Diseases and Intensive Care, Medical Faculty, RWTH Aachen University, 52074 Aachen, Germany; akoch@ukaachen.de; 3Department of General, Visceral and Transplantation Surgery, Medical Faculty, RWTH Aachen University, 52074 Aachen, Germany; aandert@ukaachen.de; 4Department of Cardiology, Nephrology and Intensive Care, Rhein-Maas Hospital, 52146 Würselen, Germany; michael.becker@rheinmaasklinikum.de; 5Department of Radiology, Medical Faculty, RWTH Aachen University, 52074 Aachen, Germany; jonae.schmoe@posteo.de (J.S.); sreinartz@ukaachen.de (S.D.R.)

**Keywords:** orthotopic liver transplantation, cardiac assessment, coronary calcium score, coronary artery disease, stress testing

## Abstract

Background: The goal of cardiac evaluation of patients awaiting orthotopic liver transplantation (OLT) is to identify the patients at risk for cardiovascular events (CVEs) in the peri- and postoperative periods by opportunistic evaluation of coronary artery calcium (CAC) in non-gated abdominal computed tomographs (CT). Methods: We hypothesized that in patients with OLT, a combination of Lee’s revised cardiac index (RCRI) and CAC scoring would improve diagnostic accuracy and prognostic impact compared to non-invasive cardiac testing. Therefore, we retrospectively evaluated 169 patients and compared prediction of CVEs by both methods. Results: Standard workup identified 22 patients with a high risk for CVEs during the transplant period, leading to coronary interventions. Eighteen patients had a CVE after transplant and a CAC score > 0. The combination of CAC and RCRI ≥ 2 had better negative (NPV) and positive predictive values (PPV) for CVEs (NPV 95.7%, PPV 81.6%) than standard non-invasive stress tests (NPV 92.0%, PPV 54.5%). Conclusion: The cutoff value of CAC > 0 by non-gated CTs combined with RCRI ≥ 2 is highly sensitive for identifying patients at risk for CVEs in the OLT population.

## 1. Introduction

Cardiovascular disease is the leading cause of non-graft-related death after orthotopic liver transplantation (OLT, LT) [1]. As patients listed for LT become older and sicker, and end-stage liver disease (ESLD) even enhances coronary artery disease (CAD), a sorrow cardiac assessment has to be performed in all patients [2,3]. Identification of an OLT patient with a coronary artery disease (CAD) is of importance for organ matching as well as for post-transplant surveillance [4,5]. The prevalence of CAD in patients considered for OLT ranges from 7.1% to 36.8% [6,7,8], mainly due to pre-existing co-morbidities. In patients with non-alcoholic steatosis hepatis (NASH), CAD prevalence rises up to 52.8% [9] due to increased inflammation. Post-transplant, the non-graft-related risk of cardiac death is about 10% [10,11]. Due to donor shortage, the use of extended criteria donor (ECD) organs had already increased up to 50% in 2012 [12,13,14]. ECD organs are associated with a significant postreperfusion syndrome [4], which leads to relevant hemodynamic compromise during the transplantation and possible functional impairment in the transplanted liver. Therefore, cardiac evaluation of the transplant recipient becomes even more important. The liver transplant recipient is especially prone to cardiovascular complications due to development or aggravation of any pre-existing metabolic syndrome such as arterial hypertension, dyslipidemia, obesity, renal injury, etc. Post-transplant indispensable immunosuppressive therapy even enhances any pre-existing metabolic syndrome and thus CAD [15,16].

There is no consensus on the preferred screening test to detect CAD in the pre-LT population nor for post-LT cardiac evaluation. The methods of cardiac assessment differ from site to site [3,8,17]. The American Heart Association and the American College of Cardiology Foundation suggest non-invasive stress testing following cardiovascular risk assessment for patients without active cardiac disease [18]. The American Study of Liver Diseases and the American Society of Transplantation advocate for invasive cardiac testing for all adult patients before OLT [19]. Cardiac catheterization has an increased risk in end-stage liver disease (ESLD) patients due to their altered coagulability [20,21].

Non-invasive testing for CAD may be performed by dobutamine stress echocardiography (DES) [22], cardiac single-photon emission computed tomography (SPECT) [23] or ECG-synchronized coronary computed tomography angiography (CCTA) [24,25]. Many patients with ESLD are incapable of performing activity to diagnose any cardiac symptoms due to the impairment of their aerobic capacity [26]. Detection of coronary artery calcification (CAC) by ECG-synchronized CT was initially introduced by Agatston [27]. Different groups verified that with the Agatston score = 0, an exclusion of CAC can be reliably detected in non-synchronized chest CT with an NPV of 91.6% [22,28]. Even non-ECG-synchronized CT scans of the abdomen in septic patients reveal the high negative predictive value of cardiovascular events by calcium scoring [29]. Each patient receives numerous CT scans of the upper abdomen during liver transplant evaluation wherein the heart is largely depicted. In these scans, the information about whether there is CAC can be easily measured without any further radiation exposure. Thus, the aim of this study was to evaluate the opportunistic benefit of measuring CAC scores from standard CT scans prior to LT by correlating with observed CVE in the postoperative period to standard cardiac assessment.

## 2. Materials and Methods

### 2.1. Patient Cohort

This retrospective cohort study was approved by the local ethics committee (University Hospital RWTH Aachen, EK 291/13). Data were collected from the electronic health records and the liver transplant database at the University Hospital Aachen. Our analyses were conducted with adherence to the Strengthening the Reporting of Observational Studies in Epidemiology (STROBE) guidelines [30].

Two hundred and seventy patients receiving an OLT in our university hospital from October 2010 to January 2018 were screened for study enrollment. Patients met the inclusion criteria if a pre-transplant evaluation included non-ECG-synchronized CT, which included scans covering the heart to a large extent. The follow-up of all patients continued until January 2019 by checking the electronical data recordings of the hospital. Patients who chose not to follow up in our hospital were asked to provide their newest medical records from their treating physician.

Overall cardiovascular risk was assessed by Lee’s revised cardiac index (RCRI) [31]. Pre-transplant, all patients received an echocardiography at rest. Patients who were not able to climb 2 stairs without dyspnea, with known coronary artery disease (CAD) or two of the following risk factors: diabetes mellitus, older than 50 years of age, smoking, hypotension or fat metabolism disorder, received either a dobutamine stress echo (DSE) [32], a single-photon emission computed tomography (SPECT) [24,26], a magnetic resonance imaging (MRI) [32,33] of the heart or an exercise ECG (ergometry). In cases of symptomatic CAD, as well as of a positive non-invasive test, an invasive coronary angiography (ICA) [32] was performed.

### 2.2. Definition of Endpoints

The primary endpoint of this study was the occurrence of cardiovascular events (CVE) following OLT. CVE is defined as acute coronary syndrome, necessity of coronary revascularization or cardiopulmonary resuscitation. Death was not included as cause of death was frequently unclear. Therefore, the aim of this study was to describe the predictive value of CAC scores obtained from non-ECG-synchronized CT for CVE and compare them to RCRI ≥ 2 and pathologic non-invasive stress testing of the heart.

### 2.3. Coronary Artery Calcium (CAC) Scoring

The quantitative evaluation of coronary artery calcifications (CAC) was performed using non-ECG-synchronized CT scans. These were non-contrast or contrast-enhanced examinations of the abdomen or liver that were performed as part of the regular pre-OLT work-up [34,35].

Normally, this protocol consists of a native, arterial, portal venous and a late phase of the venous contrast medium and routinely covers large parts of the heart. When the native examination has been omitted, the venous contrast agent phase or the phase with the biggest heart coverage is preferred for measuring CAC.

In case of absence of the native phase and due to vascular attenuation by the contrast medium, the threshold value for identifying calcium was chosen to be two standard deviations above the vascular mean value determined by ROI (Region of Interest) measurement. The standard reconstruction kernel (B30f) and a slice with 5 mm thickness were selected. Two experienced readers quantified, in consensus, the resulting Agatston and volume scores for all coronary vessels and aortic valve leaflets.

### 2.4. Statistical Analysis

The differences between groups for parametric data were analyzed by t-test or Mann–Whitney U test and displayed by mean and standard deviation or range (minimum–maximum). Nominal or ordinal scaled data were investigated for significant differences between groups by the chi-square test and reported as absolute and relative occurrence (SPSS 24.0, IBM statistics, IBM, Ehningen, Germany). The sensitivity, specificity, positive predictive value (PPV), negative predictive value (NPV) and accuracy of each test to predict CVE were calculated according to the usual definitions. Furthermore, only the combination of an RCRI ≥ 2 and CAC > 0 was of interest as they require no additional testing. The McNemar test was used to compare the sensitivities, specificities and diagnostic accuracy of the different scores or tests.

## 3. Results

One hundred and sixty-nine patients met the inclusion criteria of non-ECG-synchronized CT scans including large parts of the heart. During the follow-up period (1797 days (MW), range 1–3268) after OLT, 18/169 (10.7%) patients showed a CVE. Patients with CVE were older and more often male, but they had an equal BMI, labMELD and distribution of underlying cause of liver disease. There were no significant differences in left ventricular function (LVEF) pre-transplant. Patients were more often already diagnosed with hypertension and diabetes before OLT. Their diagnosis of CAD was also reflected by their differences in medication by aspirin, ß-blockers and statins (see Table 1). Two or more risk factors were reported in patients 49/169 (29%) of which 11 (28.2%) showed a CVE post OLT. The PPV and the NPV for two or more RCRI to detect patients for CVE after OLT were 77.6% and 94.2%. Additional data are given in Table 2.

For 111 patients, the extended pre-transplant cardio-workup included non-invasive cardiac stress testing. Methods for non-invasive cardiac stress testing differed due to the availability of the testing method as well as the tolerance of the patient for the respective exercise testing facility and to the adverse effects of the different pharmacologic agents. DSE was the most frequent test, followed by SPECT and MRI. The fewest number of patients (5/111) passed an ergometric stress test. SPECT demonstrated the highest rate of pathologic results (*p* = 0.003). Fifty-eight patients were transplanted without a stress test before OLT. The reasons for missing stress tests are not comprehensible in this retrospective analysis (e.g., patients on the ICU ward before OLT, timely matters before OLT). The distribution of patients having a stress test before OLT showing a CVE post-OLT (29/111, 26%) is comparable to the number of patients without a stress test having a CVE (16/58,18%) (see Table 3).

Due to our pre-transplant cardio-workup, 13/111 patients were identified with a suspected obstructive CAD following non-invasive stress testing. Nine patients were scheduled for invasive testing without a prior non-invasive stress test, due to being highly suspicious for obstructive coronary artery disease (medical history) as well as due to contraindications for non-invasive stress testing. Twenty-two patients received an invasive coronary angiography (ICA). In summary, invasive testing was more frequent in patients with CVE after transplant. In 3/22 patients receiving an ICA, a percutaneous coronary intervention was performed without repeating the stress test prior to OLT. In the follow-up period, only one of these patients suffered from a non-fatal NSTEMI, but two died due to progressing LV-insufficiency combined with ARDS and cerebral abscesses, respectively.

We discriminated between patients without an Agatston score at all (CAC = 0, *n* = 114, 67.5%) and those patients with an Agatston score > 0 (CAC > 0, *n* = 55; 32.5%). More patients without CVE had CAC = 0 (*p* < 0.001), while 15/18 with CVE post-OLT demonstrated a positive non-invasive stress test or CAC > 0 (vs. 45/151; *p* < 0.001). The highest sensitivity to predict CVE post-transplant was given for a CAC > 0 (77.8%). A positive stress test showed the highest specificity to identify patients at risk for CVE post-OLT. Although, the PPV for a pathologic stress test was only 54.5% in our setting. On the contrary, the combination of two or more RCRI and CAC > 0 showed a PPV of 81.6% for CVE post-OLT. The negative predictive values for stress testing (NPV = 92.0%) were comparable to a CAC = 0 and RCRI ≥ 2 (NPV = 96.5%).

## 4. Discussion

In this single-center retrospective analysis of 169 OLT recipients, we found a 10.7% incidence of CVE post-OLT. This occurrence is comparable to published data [36]. We demonstrated in our cohort that CVE post-OLT could be best predicted by the combination of RCRI ≥ 2 and CAC > 0.

There are two different etiologies for perioperative myocardial infarction: on demand myocardial ischemia and coronary plaque rupture [37]. Non-invasive stress testing methods, which are aiming for myocardial wall motion (DSE) disturbances and perfusion/glucose utilization (MRI, SPECT) as well as the RCRI, may depict patients susceptible to demand myocardial ischemia. Imaging modalities addressing coronary plaque visualization (CCTA) may detect culprit lesions that lead to acute coronary syndromes (napkin-ring sign, low attenuation plaque) [38]. Furthermore, CCTA provides a high negative predictive value for excluding significant CAD [39].

In OLT patients, the diagnostic accuracy for obstructive coronary artery disease by non-invasive stress testing, as well as of invasive cardiac diagnostic by ICA, is discussed. In a comprehensive systematic review, Konerman et al. reported the striking variability regarding sensitivity for cardiovascular events by different cardiac imaging modalities pre-OLT [36]. DSE might result in false negative results due to altered hemodynamics in ESLD. For example, high-output failure, cirrhotic cardiomyopathy and use of beta-blockers, as well as anemia, may disguise coronary artery disease in ESLD patients [2,40]. This may result in a preterm interruption of DSE due to global cardiac insufficiency. DSE has been shown to have a low sensitivity (32%) and PPV (37%) in predicting CAD [41] in the OLT population [40]. SPECT imaging is even an inaccurate screening test in ESLD patients with low sensitivity of 35% due to endothelial dysfunction in liver cirrhosis [41]. Cardiac MRI has gained increasing attention in pre-OLT screening as the morphology of the left and right ventricle can be equally assessed by an independent observer [42,43]. Reddy et al. described a sensitivity of 50% to detect CAD by MRI and a specifity of 98% [42]. Wray et al. showed the high diagnostic accuracy for CAD by ICA in OLT patients [6]. Although we only had the small number of two patients receiving a PCI pre-transplant, PCI could not prevent CVE after transplant in these patients. Two out of three patients showed a CVE after the transplantation. Wray et al. described a trend of a higher mortality in patients with significant CAD and PCI [6] as periprocedural risk increases due to the altered state of coagulability in ESLD. ICA is highly sensitive for diagnosing stenotic CAD. Hence, the consistency of the coronary plaque structure, and therefore the vulnerability of this plaque, is of further interest. Imaging modalities such as DSE, SPECT and MRI are able to identify the patient with symptomatic CAD due to demand myocardial ischemia [39,44]. One of the advantages of CCTA, besides being the benchmark for chronic coronary syndrome patients [39], is the non-invasive detection of coronary artery stenosis and the vulnerability of the underlying plaque [45]. Still, these data have to be verified in this special cohort of OLT patients.

The 2019 guidelines of the European Society of Cardiologists implemented CAC scoring to determine the clinical likelihood of obstructive CAD in pre-operative work-up for major non-cardiac surgery [39]. CAD is predictable by screening for CAC even in non-gated chest CTs [6]. Our study demonstrated the high negative predictive value of CAC (96.5%) discovered on non-ECG-synchronized CTs in these patients. Our data outgo the reported negative predictive value (91.6%) of Agatston = 0 for CAD in non-gated thorax CT by Azour et al. [22].

Joshi et al. as well as Blaha et al. described comparable findings that a CAC = 0 has a high 10-year diagnostic screening role [46,47]. As for the patients at risk of coronary plaque rupture, additional coronary artery CT angiography may help to identify these lesions [48].

Our results suggest that CAC scoring and evaluation of RCRI may reduce the number of non-invasive stress tests and ICAs. Following our results published by Mechelink et al., echocardiography at rest with strain analysis in combination with CAC scoring as well as RCRI will even help to improve the identification of patients at risk [49].

All of the 169 OLT patients reported in this study were treated in the same center under standardized procedures pre-transplant, intraoperatively and post-transplant [4,50]. Therefore, we achieved a homogeneous single-center study sample, though a single-center study limits the external validity of the results. In Germany in 2019, 22 transplant centers reported not less than 1571 liver grafts, with each transplant center following their own standard operating protocol. Furthermore, the partial retrospective design of our analyses could impair data quality. Systematic errors as well as any observer bias may not be excluded.

## 5. Conclusions

Determining the RCRI of a patient and CAC = 0 on routine non-gated CT will help to better identify patients with CAD at risk for a CVE after OLT. The high negative predictive value of Agatston = 0 combined with RCRI ≥ 2 will narrow down the number of patients who have to receive a full cardiac workup before OLT. This concept is particularly practical for the evaluation of highly compromised patients, for example in the intensive ward pre-transplant. Identifying patients at risk for CVE during and after OLT will enable a better organ-to-recipient match regarding the increasing number of ECD organs as well as thorough post-transplant monitoring of these patients.

## Figures and Tables

**Table 1 jcm-10-05148-t001:** Demographic and clinical characteristics of 169 single-center liver transplant recipients. Patients were discriminated between patients without a cardiovascular event (w/o CVE) and with a cardiovascular event post-transplant (with CVE).

	All *n* = 169	w/o CVE Post-Transplant *n* = 151	with CVE Post-Transplant *n* = 18	*p*
Age (years)	53.4 ± 11.7	52.8 ± 12.1	58.7 ± 6.2	0.043
Sex (female)	58 (34.3%)	50 (33.1%)	4 (22.2%)	<0.001
BMI (kg/cm^2^)	26.7 ± 5.6	26.7 ± 5.6	27.5 ± 5	0.564
labMELD	17.8 ± 8.8	18 ± 9	16.7 ± 6.9	0.562
Reason for transplantation				0.842
Alcoholic Cirrhosis	50 (26%)	41 (27.2%)	9 (50%)	
HCC	19 (11.2%)	17 (11.3%)	2 (11.1%)	
Acute Liver Failure	16 (9.5%)	15 (9.9%)	1 (5.5%)	
ACLF	5 (3%)	5 (3.3%)	0	
PSC	13 (7.7%)	12 (7.9%)	1 (5.5%)	
HBV/HCV-Cirrhosis	14 (8.3%)	12 (7.9%)	2 (11.1%)	
Graft Failure	1 (0.6%)	1 (0.6%)	0	
NASH	13 (7.7%)	12 (7.9%)	1 (5.5%)	
Other	13 (7.7%)	13 (8.6%)	1 (5.5%)	
CAC volume	76.4 ± 272.2	41.6 ± 110	367.8 ± 724.6	< 0.001
CAC mass	15.0 ± 52.6	8.0 ± 21.3	73.5 ± 138.9	< 0.001
Agatston score	82.6 ± 317.1	41.7 ± 113.7	425.0 ± 860.7	< 0.001
Agatston score = 0	114 (67.5%)	110 (72.8%)	4 (2.2%)	
Diabetes	54 (32%)	42 (27.8%)	12 (66.7%)	0.001
Hypertension	58 (34.3%)	47 (31.1%)	11 (61.1%)	0.011
Renal insufficiency	53 (31.4)	47 (31.1%)	6 (59%)	0.849
TIA/apoplex	3 (1.8%)	2 (1.3%)	1 (5.6%)	0.199
Statins	19 (11.2%)	13 (8.6%)	6 (33.3%)	0.002
ß blocker	18 (10.7%)	77 (51%)	11 (61%)	0.417
ASS	14 (8.3%)	12 (7.9%)	2 (11.1%)	0.645
LV ejection fraction (%)	61.0 ± 6.6	60.8 ± 6.7	62.1 ± 6.2	0.622
Smoking	30 (17.8)	24 (16%)	6 (33.3%)	0.188
known CAD	18	11 (7.2%)	7 (3.8%)	<0.001
Pathologic Stress Test	13	8	5	<0.001
rCRI (*n*)				0.002
1	120	113 (94.2%)	7 (5.8%)	
2	38	31 (81.6%)	7 (18.4%)	
3	11	7 (63.6%)	4 (36.4%)	
ICA (*n*)	22	16	6	0.007
With PCI (*n*)	3	1	2	0.001

BMI: body mass index, rCRI: Lee’s revised cardiac index, CAD: coronary artery disease, DSE: dobutamine stress echo, SPECT: single-photon emission tomography, ICA: invasive coronary angiography, PCI: percutaneous coronary intervention, LV ejection fraction: left ventricular ejection fraction, MRI: magnetic resonance imaging, Ergo: ergometry, HCC: hepatocellular carcinoma, ACLF: acute-on-chronic liver failure, PSC: primary sclerosing cholangitis, HBV: hepatitis B virus, HCV: hepatitis C virus, NASH: non-alcoholic steatosis hepatis, labMELD: laboratory model of end-stage liver disease, MACE: major adverse cardiac events, ICA: invasive coronary angiography.

**Table 2 jcm-10-05148-t002:** Accuracy of the stress tests and Agatston score to detect obstructive coronary artery disease.

	Pathol. Stress-Test	RCRI ≥2	Agatston > 0	RCRI ≥ 2 and Agatston > 0
Sensitivity	38.5	61.1	77.8	77.8
Specificity	93.9	74.8	72.8	58.9
PPV	54.5	77.6	74.5	81.6
NPV	92.0	94.2	96.5	95.7
Accuracy	87.4	73.4	73.4	60.9
*p*-Values				
vs. stress test		<0.001	<0.001	0.041
vs. RCRI			0.470	<0.001
vs. Agatston > 0				<0.001

rCRI: Lee’s revised cardiac index, PPV: positive predictive value, NPV: negative predictive value.

**Table 3 jcm-10-05148-t003:** Number of patients with and without a non-invasive stress test and their respective amount of CVE.

		CVE		Total
		no	yes	
non-invasive stress test	normal	77	23 (23%)	100
	pathol.	5	6 (55%)	11
	total	82	29 (26%)	111
	no test	42	16 (18%)	58
total		124	45 (27%)	169

## Data Availability

The datasets generated and/or analyzed during the current study are not publicly available due to German Data Protection laws but are available from the corresponding author on reasonable request after approval of the local ethics committee and data safety board.

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
