# Peer review of "Cardiovascular Evaluation of Liver Transplant Patients by Using Coronary Calcium Scoring in ECG-Synchronized Computed Tomographic Scans"

_jcm, 2021, doi:10.3390/jcm10215148_

Round 1
Reviewer 1 Report
Roehl and Colleagues conducted a retrospective study regarding 169 patients who received orthotopic liver transplantation (OLT) and who underwent non-gated abdominal computed tomography (CT) during pre-transplant evaluation. They evaluated the occurrence of cardiovascular events (the leading cause of non-graft related death in these patients) following OLT. The aim of the study was to identify the predictive value of coronary artery calcification score obtained by abdominal CT scan and compare them to revised Lee’s score and/or traditional cardiac stress tests.
Out of 169 patients, 110 underwent non-invasive cardiac stress test and 22 received an invasive coronary angiography. 3/22 received a percutaneous coronary intervention. 18/169 showed cardiovascular event after OLT. They were older, more often male with hypertension or diabetes and with significantly increased CAC mass and volume at CT scan. Patient with CAC 0 had no CVE during follow up, 15/18 with CVE post OLT had a positive non-invasive stress test or CAC > 0.
This interesting study supports the use of CAC score derived by abdominal CT scan as tool to stratify the risk of cardiovascular events after OLT. In this field, no more (cardiac) CT scan should be performed in the pre-transplant evaluation.
Some minor revisions need to be done.
1- Abstract, line 25 “ CAD and RCRI>2”, please replace CaD with CAC
2- Table 1: some percentages are missing in parentheses
3- Table 2: acronyms and abbreviations are missing in the caption
4- 59 patients did not receive an imaging test. How many of these patients had a CVE after OLT? How was the CAC score of these patients? Please add a graphic in which patients are divided into 2 groups according by imaging test (110 vs 59) and report how many CVE post OLT occurred in each group.
5- Did patients repeat a cardiovascular stress test after angioplasty and before OLT? Did these 3 patients have any post-transplant events? Please specify in the results section.
Author Response
Dear Rewiever,
thank you very much for your detailed revision. We responded to your remarks point by point in an extra document. The paper was initially given to the American Journal Experts for English editing. We gave the language and spell check a deep look.
Sincerely
Anna Röhl

Reviewer 2 Report
The authors present an important problem of the cardiovascular evaluation of patients awaiting orthotopic liver transplantation. The introduction proveides sufficient informations about the analysed problems. The patients cohort and methods are adequately described. The results are clearly presented. The conclusions are interesting from the practical point of view.
Author Response
Dear Reviewer, thank you very much for your review. We gave the manuscript initially to the American Journal Experts for english editing. We took another deep look into the englisch wording and spell ceck.
Sincerely
Anna Röhl